# Genetic Influence on the Peripheral Differentiation Signature of Vδ2+ γδ and CD4+ αβ T Cells in Adults

**DOI:** 10.3390/cells10020373

**Published:** 2021-02-11

**Authors:** Nicola Beucke, Svenja Wingerter, Karin Hähnel, Lisbeth Aagaard Larsen, Kaare Christensen, Graham Pawelec, Kilian Wistuba-Hamprecht

**Affiliations:** 1Department of Dermatology, University Medical Center, 72072 Tübingen, Germany; nicola.beucke@student.uni-tuebingen.de (N.B.); svenja.wingerter@student.uni-tuebingen.de (S.W.); 2Department of Internal Medicine II, University Hospital Tübingen, 72076 Tübingen, Germany; Karin.Haehnel@med.uni-tuebingen.de; 3The Danish Twin Register, University of Southern Denmark, 5000 Odense C, Denmark; laalarsen@health.sdu.dk (L.A.L.); KChristensen@health.sdu.dk (K.C.); 4Interfaculty Institute for Cell Biology, Department of Immunology, University of Tübingen, 72076 Tübingen, Germany; graham.pawelec@uni-tuebingen.de; 5Health Sciences North Research Institute, Sudbury, ON P3E 2H2, Canada

**Keywords:** γδ T cell differentiation, genetic impact, CMV, flow cytometry, immunomonitoring

## Abstract

Adaptive as well as innate immune traits are variously affected by environmental and genetic influences, but little is known about the impact of genetics on the diversity, differentiation and functionality of γδ T cells in humans. Here, we analyzed a cohort of 95 middle-aged twins from the Danish Twin Registry. The differentiation status of peripheral αβ and γδ T cells was assessed by flow cytometry based on the surface expression of CD27, CD28 and CD45RA. Our data confirm the established associations of latent cytomegalovirus (CMV) infection with an accumulation of late differentiated memory T cells in the αβ compartment as well as in the Vδ1+ γδ T cell subset. A comparison of differentiation phenotypes of γδ and αβ T cells that were not affected by CMV seropositivity identified a significant correlation of early differentiated (ED) Vδ2+ and intermediate differentiated (ID) CD4+ T cells in monozygotic (MZ), but not in dizygotic (DZ) co-twins. Thus, our data suggest a genetic influence on the differentiation of γδ and αß T cell subsets.

## 1. Introduction

The distribution of peripheral blood T cell differentiation signatures provides biomarker information on the capability of the immune system to react to newly encountered challenges or chronic exposures. A major physiological factor that shapes this individual T cell differentiation signature is aging—a process that is accompanied by dysregulation as well as adaptive remodeling of immune responses [1,2]. Characteristic features of older adults include a higher proportion of memory T cells and the developmentally programmed thymic involution leading to a marked reduction in the numbers of naïve T cells released to the periphery [3]. Importantly, similar shifts in the T cell differentiation signature are also associated with a latent cytomegalovirus (CMV) infection [4]. This widespread β-herpesvirus usually causes an asymptomatic infection, but can be life-threatening for newborns or immunocompromised individuals [5], and its impact on later healthspan and lifespan is controversial. The majority of individuals with such a latent infection presents an expanded CD8+ memory subset and a small late differentiated CD4+ memory subset [6], suggesting the presence of CMV as a major contributor to the phenotype of immunosenescence.

The well-studied effects of aging and CMV on the differentiation signature of “classical” CD4+ and CD8+ αβ T cells can also be seen in the rather “unconventional” γδ T cells. These represent a numerically minor population in the peripheral blood of healthy adults, accounting for 1–10% of all T cells. While the majority of γδ T cells in the peripheral blood is usually represented by the Vδ2+ subset, Vδ1+ γδ T cells are often a minor subset in the periphery but predominate in tissues such as the large intestine [7]. In contrast to αβ T cells, γδ T cells are not MHC-restricted [8] and so far only a few TCR ligands have been described, including stress-induced self-antigens [9,10]. Different γδ subsets play multiple roles in immune defense and are involved in tissue surveillance, tumor immunity, anti-microbial responses and autoimmunity [11,12]. In addition to cytokine secretion and cytotoxic activity, γδ T cells can interact with B cells, e.g., to promote immunoglobulin class switching [13,14], and are able to present antigens in an MHC-restricted fashion to αβ T cells [15]. Furthermore, CD16-expressing γδ T cells mediate antibody-dependent cellular cytotoxicity (ADCC) [16]. Vδ2+ γδ T cells usually represent the main subset of circulating γδ T cells in humans. The majority of Vδ2+ γδ T cells co-expresses the Vγ9 TCR chain, displays innate-like functions, responds to phosphoantigens and expresses semi-invariant TCRs [17]. Vδ1+ γδ T cells, on the other hand, behave predominantly like adaptive immune cells and exhibit an often private, initially clonally heterogeneous repertoire that shows marked clonal selection during childhood [18]. While Vδ2+ γδ T cells are mainly associated with the immune response against bacterial infections, Vδ1+ γδ T cells are involved in anti-viral immunity and both subsets play a role in tumor immunity [19].

The differentiation status of αβ T cells can be assessed through the pattern of expression of different isoforms of the CD45 protein tyrosine phosphatase receptor type C in combination with the co-stimulatory receptors CD27 and CD28. This model has also been used for the identification of early, intermediate and late differentiated γδ T cell subsets [6,20,21,22,23,24], although the functionality of some γδ T cell subsets might diverge from the characteristics of this adapted model [25]. Vδ2+ γδ T cells usually present a heterogeneous pattern of various differentiation stages, while the Vδ1+ γδ T cell subset is often dominated by late differentiated phenotypes in adults [22,24]. We previously found a prominent accumulation of such late differentiated Vδ1+ T cells in CMV-seropositive compared to CMV-seronegative healthy adults [6,21], as well as in stage IV melanoma patients with poor prognosis compared to those with a superior prognosis [22] [Wistuba-Hamprecht et. al, unpublished data], implying that this subset might also be contributing to immunosenescence.

Little is known about the impact of genetic background on naïve T cell repertoires and environmental factors that might influence shaping of peripheral T cell signatures. Microbial exposure after birth has recently been suggested as a driver for polyclonal expansion of public Vδ2 clones implying a major influence of the external environment [26], but not much is known about genetic associations with the control of γδ T cell distribution. However, we have previously reported correlations of CD4+ and CD8+ T cell differentiation indices in CMV-seropositive monozygotic (MZ) but not in dizygotic (DZ) twin pairs [27], consistent with a strong genetic influence. Here, we studied the genetic impact on the “unconventional” γδ T cell compartment in comparison to the “classical” CD4+ and CD8+ T cells in a subsample of the same cohort [27] of middle-aged MZ and DZ twin pairs from the Danish Twin Registry [28]. Significantly greater similarity between MZ twins, who share all their genes, compared to DZ twins, sharing approximately half of their genes, implies the existence of a genetic influence on the investigated trait. Here, we show a correlation in the distribution of Vδ2+ γδ T cells with an early differentiation signature and intermediate differentiated CD4+ T cells in MZ, but not in DZ twins.

## 2. Material and Methods

### 2.1. Cohort

The studied cohort of ninety-five individuals (Table 1) was recruited from the 2008–2011 survey of middle-aged twins of the Danish Twin Registry [28] based on sample availability and overlaps with the cohort of the earlier study [27]. Zygosity was determined by a questionnaire on the degree of similarity between twins in a pair [29] and only twin pairs of the same sex were included. Samples were collected from 2008–2011 using Vacutainer CPT tubes (BD) for blood draws, and peripheral blood mononuclear cells (PBMCs) were immediately isolated according to the manufacturer’s protocol, cryopreserved and stored in liquid nitrogen. The anti-CMV IgG serostatus was determined using a recombinant CMV IgG immunoblot kit (Mikrogen) detecting six viral target molecules (IE-1, p150, CM2, p65, gB1 and gB2). The study was approved by the Science Ethics Committee of Southern Denmark (S-VF-19980072).

### 2.2. Flow Cytometry

Phenotypic analysis followed standardized protocols on cryopreserved samples. The latter were thawed and stained with monoclonal antibodies (Appendix A) according to our previously published OMIP-20 panel [20]. All antibodies were titrated to determine the optimal concentration in the orchestration of the entire panel. Briefly, cells were incubated with an Fc-receptor-blocking reagent (Gammunex), and ethidium monoazide bromide (EMA) was used to exclude dead cells. CD3+ T cells were divided into γδ TCR+/− populations. The γδ TCR+ compartment was subdivided based on expression of Vδ1 and Vδ2 whereas the γδ TCR– population was subdivided into CD4+ and CD8+ T cells. Differentiation phenotypes were assessed by staining of CD27, CD28 and CD45RA. CD16 expression was studied to identify potentially ADCC-associated T cell subsets. A three laser LSR-II flow cytometer (BD) with customized filter settings (Appendix A), running on FACSDiva software V6.1.3 (BD), was used for data acquisition. Single color controls were used for automatic compensation, and cells from a single batch of PBMCs from the same donor were included in each run to ensure comparability between different days. Data analysis was performed using FlowJo V10.5.3 (BD) following the gating strategy displayed in Appendix A. Populations with less than 120 events were excluded from further subset analysis.

### 2.3. Statistics

Statistical analysis was performed with Prism V5.04 (GraphPad). The cohort was divided based on zygosity, CMV serostatus or mean Vδ1 frequency. Population frequencies between groups were compared using the non-parametric Mann–Whitney-U test. Correlations between twin pairs were analyzed using non-parametric Spearman testing with a defined cut-off of r > ±0.8 to select only strong correlations for further analyses. The one-way random intraclass correlation coefficient (ICC) was calculated using SPSS V24 (IBM). Similar to the previous study of Goldeck et al. [27], standard biometrical heritability analyses were performed to estimate the relative contribution of genetic and environmental factors. The total phenotypic variance can be divided into additive genetic (A), dominance genetic (D), shared environmental (C) and nonshared environmental (E) effects. The best fitting and simplest models were selected. The broad-sense heritability describes the proportion of the total phenotypic variance due to genetic variance. Heritability analyses were performed without adjustments and also adjusting for age and gender with and without CMV serostatus for CD4+ CD27+CD28+CD45RA– and Vδ2+ CD27+CD28+CD45RA+ T cells using the subsample of intact twin pairs only. The statistical program R v3.5.2 and the Mets package Analysis of Multivariate Event Times v1.2.6 were used. 

*p*-values < 0.05 were classified as statistically significant throughout the entire study. Where applicable, the Bonferroni correction for multiple comparisons was used.

## 3. Results

### 3.1. Definition of a T Cell Differentiation Signature

Detailed phenotyping of peripheral γδ and αβ T cells was performed in a cohort of 95 middle aged members from the Danish Twin Registry (Table 1). Within the whole CD3+ T cell population, we measured mean frequencies of 1.4% and 1.7% for Vδ1+ and Vδ2+ γδ T cells, respectively. The mean frequencies in the αβ T cell compartment were 63.1% for CD4+ and 27.1% for CD8+ T cells within the CD3+ T cell population (Figure 1, upper part). Based on expression of CD27, CD28 and CD45RA we identified several differentiation phenotypes in both the γδ and αβ T cell subsets. Early differentiated (ED) T cells were classified as CD27+CD28+CD45RA+, intermediate differentiated (ID) T cells as CD27+CD28+CD45RA– whereas CD27–CD28–CD45RA+ T cells were considered very late or terminally differentiated (LD). The distribution of memory differentiation phenotypes is summarized in Figure 1 (lower part). Each horizontal row in the four heat map blocks (Vδ1+, Vδ2+, CD4+ and CD8+) represents a single individual. Each column presents the differentiation phenotype assessed by CD27, CD28 and CD45RA expression. Frequencies of the respective phenotypes are color coded and subjects are stratified by zygosity and CMV serostatus. The data displayed in the heatmap confirms the expected characteristics consistent with consensus findings in the literature, as well as our own findings, such as domination of the Vδ1+ subset by a LD phenotype. In contrast, the Vδ2+ subset showed a heterogeneous distribution of these differentiation phenotypes as well as high donor heterogeneity (Appendix A). In CD4+ T cells we found a dominance of ED and ID cells, whereas CD8+ T cells showed a more heterogeneous differentiation signature with high frequencies of ED T cells. 

### 3.2. Consideration of CMV as Confounder for T Cell Differentiation Signature Analyses

Consistent with previous studies [30,31,32,33] we found higher frequencies of LD and lower frequencies of ED T cells in CMV+ compared to CMV−individuals (Figure 1 lower part and Figure 2). Notably, CMV+ subjects had significantly higher frequencies of LD T cell phenotypes in the Vδ1+, CD8+ and CD4+ compartments in comparison to CMV– individuals (*p* < 0.001 for all). Significantly lower proportions of ED T cell phenotypes were observed for Vδ1+ and CD8+ T cells in CMV+ compared to CMV– individuals (*p* < 0.0001, *p* = 0.009, respectively). Furthermore, CMV seemed to influence the distribution of effector phenotypes: We identified significantly higher frequencies of ID, CD27+CD28– and CD27–CD28+ Vδ1+ T cell phenotypes in CMV– subjects (*p* < 0.0001, *p* < 0.0001 and *p* = 0.0017, respectively). Moreover, individuals with high Vδ1 frequencies (above the mean) had significantly higher proportions of LD and reciprocally lower proportions of ED Vδ1+ T cells (Appendix A). Conversely, we observed significantly lower proportions of CD27–CD28+ in the αβ compartment of CMV– individuals (CD4: *p* = 0.0029; CD8: *p* = 0.0004). In addition, the differentiated CD4+ CD27–CD28–CD45RA– phenotype had a lower abundance in CMV– subjects (*p* < 0.0001). In contrast to these findings, we detected no significant differences in the Vδ2+ compartment and in ED and ID differentiated CD4 phenotypes depending on CMV serostatus. 

Additionally, γδ and αβ T cell compartments were analyzed for the expression of the FcγRIII (CD16). In general, we found higher frequencies of CD16+ cells in γδ T cells compared to αβ T cells, and CMV– subjects showed a significantly higher frequency of CD16+ Vδ1+ cells (Appendix A).

### 3.3. Genetic Impact on the Differentiation Signature of Vδ2+ and CD4+ T Cells 

To investigate genetic influences on the peripheral T cell differentiation signature, we compared the individual differentiation signatures between MZ and DZ pairs. The distribution of ED, ID and LD T cells varied greatly between the different γδ and αβ T cell subsets (Figure 3) as well as between different individuals and was strongly influenced by CMV. While there was dominance of ED cells in the CD4+ and LD cells in the Vδ1+ compartment, CD8+ and Vδ2+ T cells showed a heterogeneous differentiation signature. Nevertheless, as expected, there were no significant differences for any investigated cellular phenotype between the groups of MZ and DZ twins (Figure 3, Appendix A). We also observed no differences in CD16 expression between these two groups (Appendix A). Furthermore, CMV infection had neither a significant impact on the Vδ2+ differentiation profile nor on ED and ID CD4 T cell phenotypes (Figure 2B,C). Thus, we assumed for the following analyses of these cellular subsets that CMV-associated effects are controlled for and that both groups (MZ and DZ) are similar overall. 

Next, we tested the cohort for intact twin pair (Appendix A) correlations of T cell differentiation phenotypes (Appendix A); only correlations with a Spearman r > ±0.8 were considered for further analysis. Between the twins in each pair, we observed a strong correlation of ED cells in the Vδ2+ compartment of MZ but not DZ twins (r = 0.832, ICC = 0.856 vs. r = 0.103, ICC = 0.207; Figure 4A,B and Table 2). Interestingly, frequencies of ID CD4+ T cells strongly correlate in MZ but not in DZ twin pairs (r = 0.870, ICC = 0.859 vs. r = 0.054, ICC = 0.154; Figure 4C,D and Table 2). No such correlations were found for ED Vδ1+ or ID CD8+ T cells (Appendix A) or the frequencies of the four parental T cell subsets (data not shown). Heritability analysis using the best fitting model (AE model) showed a high heritability for ED Vδ2+ (0.831; 95% CI: 0.688; 0.916) and ID differentiated CD4+ T cells (0.897; 95% CI: 0.795; 0.952), confirming these findings. Only minor differences were observed when adjusting for age, gender and CMV (Table 3).

## 4. Discussion

Here, we report the results of an immunomonitoring study of MZ and DZ twin pairs from the Danish Twin Registry that suggests a genetic influence on the peripheral T cell differentiation signature of γδ as well as αβ T cells, as determined using our OMIP-20 flow cytometry panel [20]. 

Vδ1+ γδ T cells seem to follow similar differentiation pathways as αβ T cells [25]. Davey et al. reported that the initially unfocused Vδ1 TCR repertoire is shaped by clonal selection and profound clonal expansions [18]. This process is accompanied by differentiation from a naïve to an effector phenotype and is associated with downregulation of CD27 and CD28, alteration of homing capabilities and acquisition of different functionality. These changes in repertoire diversity concurrent with phenotypic differentiation analogous to CD8+ T cells imply an adaptive form of immune surveillance [18]. However, this may not be the case for Vδ2+ γδ T cells, because the presence of these markers could have different implications [25]. Data suggest that this compartment comprises a major innate-like Vγ9+Vδ2+ subset and a minor Vγ9–Vδ2+ subset, with the latter showing features of adaptive immunity and behaving similarly to Vδ1+ T cells [17]. Nevertheless, this Vγ9–Vδ2+ subset is only present at low frequencies [17,34], and the Vδ2 antibody (clone B6) that we used in this study was described by Davey et al. as not suitable to detect Vγ9–Vδ2+ γδ T cells [17]. Hence, we conclude that the investigated Vδ2+ subset in our study was limited to the rather innate-like Vγ9+Vδ2+ phenotype and that our results might not apply to the Vγ9–Vδ2+ subset. While the Vδ1 TCR repertoire is private and includes complex CDR3 sequences, the Vγ9Vδ2 TCR repertoire is semi-invariant, contains public clones, and CDR3 lengths are very restricted [17,18]. Although Vγ9+Vδ2+ γδ T cells undergo rapid expansion in the first few weeks of life, these expansions are polyclonal [26,35], and neonatal and adult TCR repertoires are broadly similar regarding the degree of clonotypic focusing [17]. Furthermore, Vδ2+ γδ T cells seem to follow a different path during aging compared to αβ T cells [36,37,38] and exhibit a heterogeneous but relatively stable distribution of differentiation phenotypes over time [24,37,39]. Different Vδ2+ γδ T cells phenotypes have also been associated with differences in functionality, e.g., cells expressing CD27 showed a higher proliferative capability compared to CD27– cells, and both subsets had distinct effector potentials [39,40]. Taken together, these findings suggest a very different biology underlying Vγ9+Vδ2+ in comparison to Vδ1+ γδ T cells that needs to be addressed in future studies.

In accordance with the published literature, we confirmed an association of latent CMV infection with an accumulation of LD T cells in the αβ T cell subset [30,31,32,33] as well as in the Vδ1+ γδ T cell compartment [6,21,41], which has also been functionally linked to anti-CMV immunity [42]. While CMV seems to have a strong influence on Vδ1+ γδ T cells, only a moderate impact on Vδ2+ γδ T cells was observed, also in accordance with previous findings [24,37]. Clonal expansions and concomitant phenotypic differentiation in response to acute CMV infection have also been shown for the minor Vγ9–Vδ2+ subset [17]. Taken together, these data suggest that the Danish MZ and DZ twins studied exhibited the expected T cell phenotypes and are representative of other published cohorts. Besides CMV, age is known as a major factor influencing the T cell differentiation signature [1,2]. However, our cohort is middle-aged, the twins within a twin pair are naturally age-matched and adjustment of the heritability models for age had only a minor influence. Therefore, we assume that age-associated effects are controlled for.

Despite increasing knowledge on the differentiation pathways of γδ T cells and changes in their TCR repertoires from neonates through to aged individuals, little is known about the impact of genetics on the composition of their peripheral signatures. Only a low heritable influence was found in a twin study looking at the whole γδ T cell population [43]. This is most likely due to not distinguishing between the Vδ1+ and Vδ2+ subset. However, another study on twins described a substantial degree of environmental influence on the Vδ1+ subset, while a higher heritability of immune traits within the Vδ2+Vγ9+ subset was reported [44]. To the best of our knowledge, the present study provides a unique analysis of γδ T cells in MZ and DZ twin pairs allowing investigation of parallels and differences in their peripheral differentiation signature irrespective of environmental influences. The correlation of ED Vδ2+ T cells in MZ but not DZ twins documented here is in agreement with the previously identified higher degree of heritability in Vδ2+Vγ9+ compared to Vδ1+ γδ T cells [44] and thus the genetic impact on this γδ T cell subset. 

The close correlation of frequencies of ID CD4+ αβ T cells in MZ twin pairs, but not in DZ twins, is consistent with the results of previous investigations where we reported a correlation of a calculated differentiation index and the expression of exhaustion markers on CD4+ and CD8+ αβ T cells in the context of latent CMV infection [27]. Furthermore, in line with our data, Brodin et al. reported a high heritable influence on the frequencies of naive, central memory and CD27+ CD4+ T cells [43]. 

Nevertheless, there are limitations to our study that need to be considered. One could be that using cryopreserved PBMCs might not reflect the actual in vivo situation at the time of blood draw, for example, due to differences between different T cell subsets in their susceptibility to freezing or thawing. However, the method used here is an established approach for cohort studies like this, providing the advantage of enabling batch analysis by protocols validated in earlier studies which document a lack of problems with freezing and thawing procedures [20,27,45,46] as addressed specifically in [22]. Another limitation is the small number of twin pairs available for the present study, due to which we were not able to perform reliable analyses when stratifying according to CMV serostatus. Since the groups of MZ and DZ twin were similar overall regarding their differentiation signatures and the CMV serostatus had no significant influence on the frequencies of ED Vδ2+ and ID CD4+ T cells, we do not expect CMV to be a major confounder. Moreover, heritability analyses did not decipher confounding factors on the identified correlations between the T cell phenotype and zygosity and adjustment of the best fitting model (AE model) for CMV revealed, as expected, no impact of the latter on the identified correlations. However, it is possible that we missed correlations of those differentiation phenotypes most strongly influenced by CMV. Furthermore, it needs to be considered that the commonly applied differentiation model using the surface expression of CD27, CD28 and CD45RA, also used in this study, was initially established for αβ T cells [47,48] and might not be equally applicable to γδ T cells. This emphasizes the urgent need to study genetics and functionality of γδ T cells in greater detail to better understand their ontogeny and trajectories in the context of life long exposure to pathogens. However, the here identified genetic influence on early differentiated T cells is in line with the data from Orrù et al. [49] showing a decreasing heritability from naive via memory to terminally differentiated T cells, possibly because late differentiated subsets are more strongly influenced by environmental factors. So far, very little is known about genetic elements regulating the immune system under homeostatic conditions. Besides the regulation of immune cell frequencies in the blood by mechanisms controlling the proliferation and elimination of certain circulating subsets, regulation can also occur on the level of protein expression [50]. Moreover, one gene can control several immune traits and one immune trait can be influenced by multiple genes. Recently, several such associations between genetic loci and immune traits have been identified by genome-wide association studies (GWAS) [49,50], starting to uncover the complex mechanisms involved in genetic regulation of the immune system; further studies will elucidate genes controlling the γδ T cell population. 

Taken together, our data shows a higher degree of similarity in the frequencies of Vδ2+ γδ T cells with an early differentiation signature in middle-aged MZ twins than in DZ twins. This indicates that the differentiation signature of peripheral Vδ2+ γδ T cells is genetically influenced, over and above differences in the immune biography caused by environmental exposures over the lifetime. Whether this also applies to the γδ TCR repertoire and/or functionality as well as the underlying mechanisms of genetic regulation needs to be addressed in further investigations.

## Figures and Tables

**Figure 1 cells-10-00373-f001:**
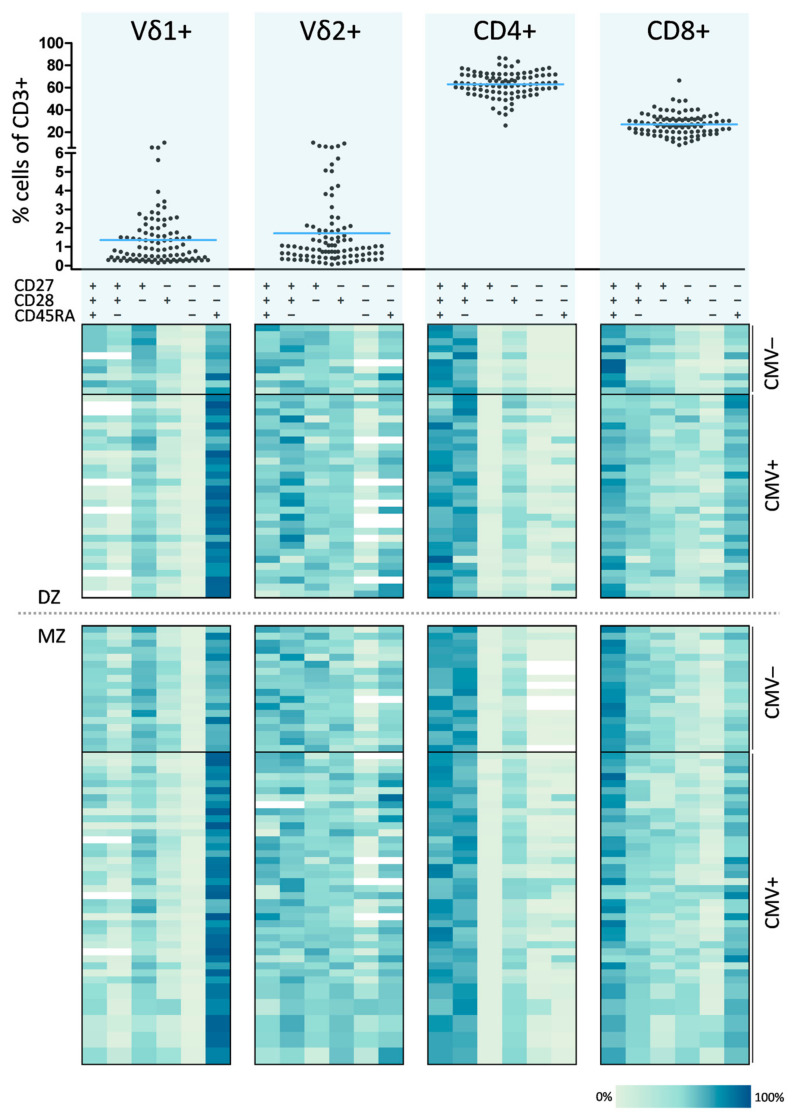
Peripheral frequencies of γδ and αβ T cell subsets and distribution of corresponding differentiation phenotypes. Frequencies of Vδ1+, Vδ2+, CD4+ and CD8+ T cells within all CD3+ T cells are shown. Blue lines indicate the mean for each group. Heat maps illustrate the differentiation signatures for each subset based on the combined expression of CD27, CD28 and CD45RA. Each line represents a single individual across all four T cell subsets, and each row presents a defined differentiation T cell phenotype. Frequencies for each phenotype are color-coded. The upper heat maps represent dizygotic (DZ), the lower ones monozygotic (MZ) twins. Within each heat map subjects are grouped by cytomegalovirus (CMV) serostatus. +, positive for the respective feature; –, negative for the respective feature.

**Figure 2 cells-10-00373-f002:**
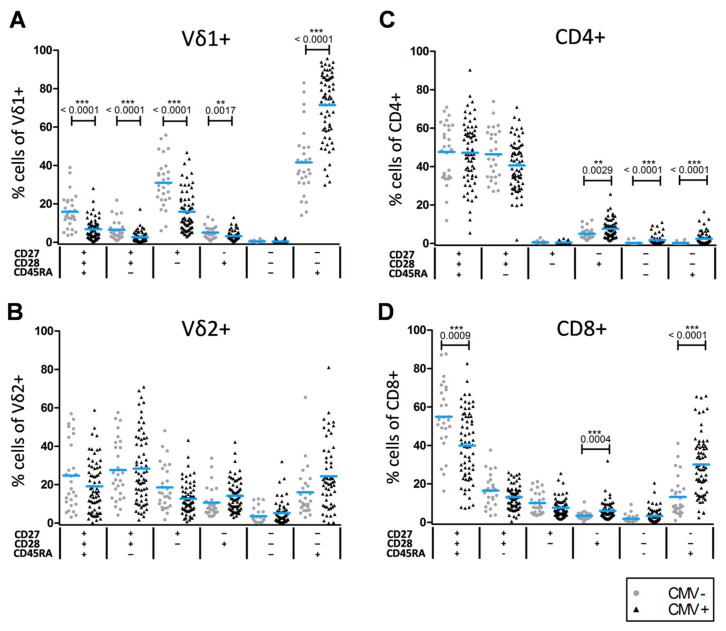
Influence of latent CMV infection on differentiation signatures of γδ and αβ T cell subsets. CMV seropositivity is associated with lower frequencies of early differentiated (ED) and/or higher frequencies of late differentiated (LD) phenotypes in the Vδ1+ (**A**), CD4+ (**C**) and CD8+ (**D**) compartments. No significant differences were observed in the Vδ2+ (**B**) compartment. Blue lines indicate the mean for each group. Statistical evaluation was performed using the Mann–Whitney-*U* test with Bonferroni correction for multiple comparisons: *p* < 0.0083 was defined as statistically significant. Non-significant results are not shown. **, *p* < 0.01; ***, *p* < 0.001; +, positive for the respective marker; –, negative for the respective marker.

**Figure 3 cells-10-00373-f003:**
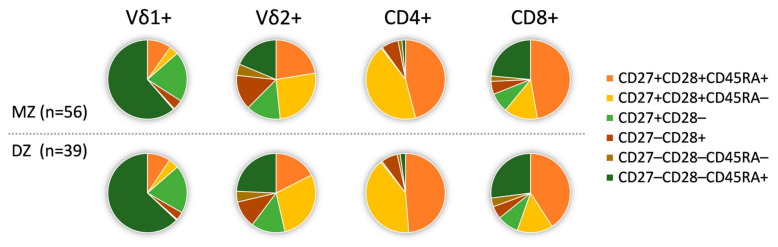
Similar differentiation status of T cell subsets in monozygotic and dizygotic twins. The distribution of differentiation phenotypes varied between Vδ1+, Vδ2+, CD4+ and CD8+ T cells, but was comparable between the groups of MZ and DZ twins. Pie charts represent the mean of each differentiation phenotype in the sub-groups of MZ and DZ twins.

**Figure 4 cells-10-00373-f004:**
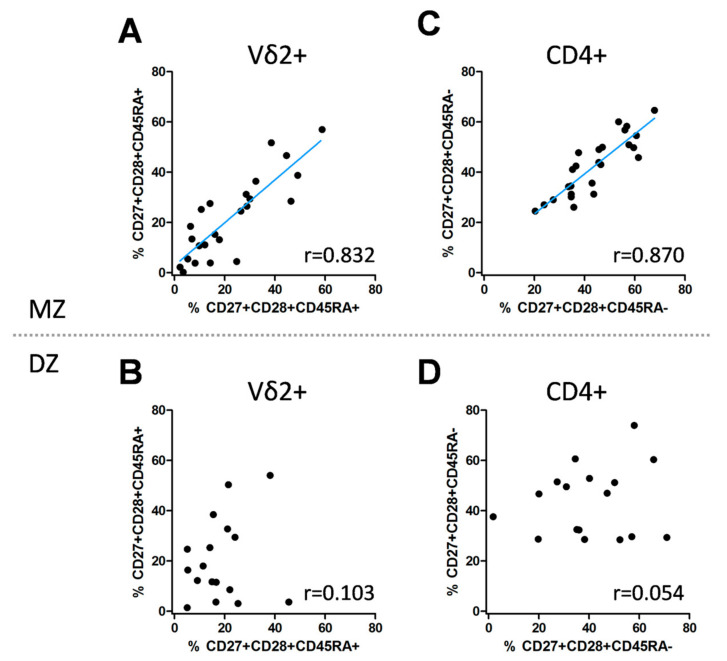
Correlation of the T cell differentiation status in monozygotic and dizygotic co-twins. CD27+CD28+CD45RA+ Vδ2+ T cells (**A**,**B**) and CD27+CD28+CD45RA– CD4+ T cells (**C**,**D**) correlated strongly in MZ but not in DZ co-twins. The Spearman correlation coefficient r as well as the intraclass correlation coefficient (ICC) and corresponding *p*-values and confidence intervals (CI) were calculated (Table 2).

**Table 1 cells-10-00373-t001:** Cohort characteristics.

Zygosity	Gender	CMV Serostatus		Age (Years)
		Positive	Negative	(Range; Median)
MZ *n* = 56	♀: *n* = 39 (69.6%)	*n* = 26 (66.7%)	*n* = 13 (33.3%)	41–66; 46
	♂: *n* = 17 (30.4%)	*n* = 12 (70.6%)	*n* = 5 (29.4%)	42–64; 45
	All	*n* = 38 (67.9%)	*n* = 18 (32.1%)	41–66; 46
DZ *n* = 39	♀: *n* = 26 (66.7%)	*n* = 19 (73.1%)	*n* = 7 (26.9%)	43–73; 49.5
	♂: *n* = 13 (33.3%)	*n* = 10 (76.9%)	*n* = 3 (23.1%)	43–77; 53
	All	*n* = 29 (74.4%)	*n* = 10 (25.6%)	43–77; 50

MZ, monozygotic; DZ, dizygotic; CMV, cytomegalovirus.

**Table 2 cells-10-00373-t002:** Spearman correlation and intraclass correlation in MZ and DZ twins.

Subset	Twins	Spearman r	*p*-Value (Spearman)	95% CI (Spearman)	ICC	*p*-Value (ICC)	95% CI (ICC)
Vδ2+, CD27+CD28+CD45RA+	MZ	0.832	<0.0001	0.638; 0.927	0.856	<0.0001	0.699; 0.935
Vδ2+, CD27+CD28+ CD45RA+	DZ	0.103	0.694	−0.410; 0.567	0.207	0.200	−0.279; 0.613
CD4+, CD27+CD28+CD45RA–	MZ	0.870	<0.0001	0.718; 0.943	0.859	<0.0001	0.709; 0.935
CD4+, CD27+CD28+CD45RA-	DZ	0.054	0.837	−0.451; 0.532	0.154	0.266	−0.328; 0.577

CI, confidence interval; ICC, intraclass correlation coefficient.

**Table 3 cells-10-00373-t003:** Broad sense heritability and 95% CI using the best fitting model (AE model).

	Vδ2+, CD27+CD28+CD45RA+	CD4+, CD27+CD28+CD45RA−
Model 1 (not adjusted)	0.831 (0.688; 0.916)	0.897 (0.795; 0.952)
Model 2 (adjusted for age, gender)	0.820 (0.668; 0.912)	0.891 (0.782; 0.949)
Model 3 (adjusted for age, gender, CMV)	0.800 (0.626; 0.905)	0.887 (0.768; 0.949)

## Data Availability

The data presented in this study are available on request from the corresponding author.

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
