# Peer review of "Genetic Influence on the Peripheral Differentiation Signature of Vδ2+ γδ and CD4+ αβ T Cells in Adults"

_cells, 2021, doi:10.3390/cells10020373_

Round 1

Reviewer 1 Report

This is a nice paper, tackling an interesting problem, this is the impact of genetics on gd T cell development. Unfortunately, the authors offer no mechanistic theory, re the influence of the genetic background, which they identify.

In either the introduction or the discussion the authors might consider that Vd1 are mainly tissue-resident, and those found in the blood may represent a subset.

Why was zygosity just checked with a questionnaire, not genetically?

In Table 1 it would be helpful to be more precise and give information on the cohort characteristics in a different way (e.g. how many female subjects were CMV postivive, and fell into which age groups). Would this influence statistic analysis?

Reviewer 2 Report

This paper nicely demonstrates a clearcut genetic influence on the differentiation stages of particular T cell populations even in adults.This is nicely demonstrated in twin studies as both members of monozygotic twins have very similar levels of naive phenotype Vd2+ cells, and memory type CD4+TCRab+ cells. These experiments are technically well performed, experimental design is perfect and all required control data are there.

Major comments:

  1. difference between monozygotic and dizygotic twins is performed by questionnaire. Should this not be done by short tandem repeat analysis? If not, the authors should show or refer to evidence that this method is appropriate.
  2. While the titel of the paper states that the main topic is the genetics of the different T cell population sizes, only 1 (small) figures and 2 (small) tables deal with the subject.All other figures are actually control data and can be put in the supplementals. On the other hand: correlation data in MZ and DZ twins are only shown for 2 populations, namely ED Vd2+ en ID CD4+ cells. Why this selection? I can understand that CMV may be interfering in the dataset if the twin pairs are discordant in CMV serostatus, but the authors show no data on that. Since all these data are available, a table should be included, similar to table 2, where all populations (see figure 1 and 2)  are included. Where CMV influences the levels of a particular population, the analysis may be confined to the CMV concordant twins.
  3. There is no data on the absolute numbers of cells of each population

Minor comments

  1. Figure S5 is not referred to in the result section
  2. Line 209-211: ID and ED is reversed
  3. I would have appreciated a paragraph of discussion on the gene(s) involved in the genetics of this trait.

Reviewer 3 Report

Supplement
1. Fig. 1 - is this subject really representative? pan-gdT staining looks good, but then the Vd1/Vd2 is rather weird. Although it is possible that Vd1 predominate, it is rarely a case. Moreover, what is that non-Vd1 and non-Vd2 population? Of course, other deltas are used, but apart from Vd3 in the liver, there are hardly noticeable in human.

Introduction
1. " Furthermore, phosphoantigens can be recognized in a butyrophilin dependent manner" - Technically speaking, Vd2 gdT cells do not recognise pAgs per se. Moreover, it is a function of Vd2 cells only. Since it is clearly mentioned within some basic Vd2 characteristics, I suggest deleting this sentence

Methods
1. When was the blood collected? How were the samples stored? What was the percentage of dead PBMCs after thawing?

Results
1. There is scarcely any difference between Vd1 and Vd2 percentage in the studied group. Usually, Vd2 subset is far more numerous and creates the majority of total gdT cells. Have you considered this to be an artefact? Maybe related to freezing and thawing?
2. Fig 3 (and possibly in other places!) please correct CD45 into CD45RA.
3. Have you tested if there are any differences between paired twins for things like pan gdT, Vd1 or Vd2 percentage?
4. Have you correlated the CMV viral load with pan gdT, Vd1 or Vd2 percentage?
5. Fig 5. please provide r coefficient for each panel
6. Have you considered assessing the clonality of TCR genes in Vd1 and Vd2 e.g. by NGS?

Discussion
1. Please clearly address the major limitations of the study.

Round 2

Reviewer 2 Report

no further comments

Author Response

Dear Reviewer 2,

thank you for the positive feedback on the revised version of our manuscript

Reviewer 3 Report

Dear Authors,

I am satisfied with your response to my previous review. Still, I have some further questions:

  1. I noticed a problem with controls - it seems that CD45RA gate is fixed at sligthly above 10^3. This should have been done with a proper FMO. Judging from supplementary figures, gating of CD45RA pos and neg may not be proper. Have you performed FMO?
  2. Pacific Blue and PacOrange may have important spectral overlap (I don't know the exact configuration of machine used in the current project - please add laser and filter configuration to the supplementary files according to MiFlowCyt: https://doi.org/10.1002/cyto.a.20623). This is of no importance to CD4 and CD8, but this may have increased background staining for gdT subsets thus again it should not be a fixed gate - especially one that looks like being first established for CD4 or CD8 cells.
  3. Similarly for CD16 expression - was FMO prepared?
  4. Lastly, a comment to your response to Rev. 2 about absolute counts. You can do it using any machine - you just need counting beads.
